



# Decadal variations in atmospheric water vapor time series estimated using ground-based GNSS

Fadwa Alshawaf, Galina Dick, Stefan Heise, Tzvetan Simeonov, Sibylle Vey, Torsten Schmidt, and Jens Wickert

GFZ German Research Centre for Geosciences, Telegrafenberg, D-14473 Potsdam, Germany

*Correspondence to:* Fadwa Alshawaf (fadwa.alshawaf@gfz-potsdam.de)

**Abstract.**

Ground-based GNSS (Global Navigation Satellite Systems) have efficiently been used since the 1990s as a meteorological observing system. Recently scientists used GNSS time series of precipitable water vapor (PWV) for climate research. In this work, we use time series from GNSS, European Center for Medium-Range Weather Forecasts Reanalysis (ERA-Interim) data, and meteorological measurements to evaluate climate evolution in Central Europe. The assessment of climate change requires monitoring of different atmospheric variables such as temperature, PWV, precipitation, and snow cover. PWV time series were obtained by three methods: 1) estimated from ground-based GNSS observations using the method of precise point positioning, 2) inferred from ERA-Interim data, and 3) determined based on daily surface measurements of temperature and relative humidity. The other variables are available from surface meteorological stations or received from ERA-Interim. The PWV trend component estimated from GNSS data strongly correlates with that estimated from the other data sets. The linear trend is estimated by straight line fitting over 30 years of seasonally-adjusted PWV time series obtained using meteorological measurements. The results show a positive trend in the PWV time series at more than 60 GNSS sites with an increase of 0.3–0.6 mm/decade. In this paper, we compare the results of three stations. The temporal increment of the PWV correlates with the temporal increase in the temperature levels.

## 1 Introduction

Water vapor is the most active atmospheric constituent that permanently affects the Earth's climate. Due to its high temporal and spatial variations, the precipitable water vapor (PWV) content in the atmosphere has to be regularly and accurately determined for meteorological and climatological purposes. PWV is the a mount of water (in millimeters) results from condensing a column of air that extends from the measurement point to altitudes of about 12 km. The water vapor resides mainly in the lowest 3 km of the atmosphere and its content generally increased with warm air temperatures. While other observation systems such as radiosondes and microwave radiometers provide PWV measurements that are limited in the temporal and spatial resolutions, ground-based GNSS can provide series of accurate PWV estimates with 15 minutes (for this work) temporal sampling at a dense GNSS network, without significant additional costs. Since Bevis et al. (1992) presented GPS as an efficient meteorological tool, GNSS data have been increasingly used for estimating atmospheric parameters, particularly precipitable





water vapor (Gendt et al., 2004; Luo et al., 2008; Jade and Vijayan, 2008; Bender et al., 2008; Alshawaf et al., 2015). GNSS-based estimates of zenith total delay or PWV have been used to improve the numerical weather prediction (NWP) models (Bock et al., 2005; Bennitt and Jupp, 2012). They have also been used to improve the performance of high-resolution atmospheric models (Pichelli et al., 2010). Besides meteorology, Elgered and Jarlemark (1998) and Gradinarsky et al. (2002) employed

GNSS estimates of PWV from 1993 to 2002 over Scandinavia for climatological research. They observed that PWV shows an increase of 1.2–2.4 mm per decade. Haas et al. (2003) used ground-based GPS, very long baseline interferometry, radiosonde, and microwave radiometer data to assess long-term trends in PWV time series over Sweden.They observed an increase of about 0.17 mm/year within the period 1980–2002. The PWV time series from ground-based GNSS and European Center for Medium-Range Weather Forecasts Reanalysis (ERA-Interim) data might show temporal inconsistencies due, for example, to hardware

replacement (Ning et al., 2016). Therefore, homogenization of the atmospheric data is indispensable for climatological research to properly estimate climatic long-term trends. Vey et al. (2009) and Ning et al. (2016) analyzed PWV time series estimated at global GNSS sites to detect and correct for inhomogeneities in the data. Reanalysis atmospheric models such as ERA-Interim have also been employed for climate research. The analysis fields are produced based on 4D-Var assimilation of regular and irregular meteorological data, including surface and upper-air atmospheric fields (Dee et al., 2011). Bengtsson

et al. (2004) observed an increasing long-term trend with a slope of 0.36 mm per decade in the water vapor data set of ERA 40 (Uppala et al., 2005). They suggested to use the reanalysis data for the analysis of climatic trends. Similar positive trends were observed (Dessler and Davis, 2010) in other reanalysis data sets such as ERA-Interim (Simmons et al., 2007) and Modern Era Retrospective-Analysis for Research and Applications (MERRA) (Suarez et al., 2008). These studies conclude that reanalysis data sets are sufficiently accurate to be used as a reference for evaluating the performance of water vapor

measurement techniques.

    Typically, climate scientists consider a period of 30 years as an appropriate time over which to average variations in weather and define climate for a particular site, as described by the world meteorological organization (WMO). Data collected and averaged or summed in some way over 30 years are referred to as climate normals. A 30 year period is recommended, as it is sufficiently long to filter out the interannual variations or anomalies, but at the same time short enough to show climatic trends.

It is then obvious that the GNSS data are still too short for estimating the correct climatic trends in this sense. The previous studies done using GNSS-based PWV time series show different trend estimates due to different time windows as well as the research region for which the trend is estimated. The the current climate normal period is calculated from 1 January 1991 to 31 December 2020 (WMO). In order to retain consistency, a normal period of 30 years was used for our research starting however from 1984. Therefore, time series of length above 30 years are inevitable to support the GNSS data. We used ERA-Interim

reanalysis and meteorological data from the German Weather Service. The former provides global PWV grids while the latter do not. However, different studies have used the dew point that is computed using surface measurements of temperature and relative humidity to approximate the column PWV (Reitan, 1963; Bolsenga, 1965; Smith, 1966; Tuller, 1977). The formulas presented to obtain the PWV from surface measurements are described in section 3. These formulas require only information that can accurately be determined on the ground. The dew-point-based PWV approximations tend to be more accurate under

stable atmospheric conditions. It is however obvious that PWV estimations based just on atmospheric conditions at Earth's





surface would not always be in complete agreement with, for example, PWV values from balloon soundings integrated through the atmosphere. We can investigate these very long time series for climate analysis, particularly when they show sufficient agreement with the GNSS data.

This paper is organized as follows. In section 2, we describe the method for PWV determination using GNSS data and the comparison with ERA-Interim. The method to compute the PWV based on surface measurements of temperature and relative humidity is described in section 3. In section 4, the time series are analyzed to extract the trend and estimate the decadal variation then the conclusions are presented.

## 2   Determination of atmospheric PWV from GNSS data

For the work presented in this paper, we used data collected in central Europe, mainly in Germany as shown in Figure 1. The research region is well covered by 278 permanent GNSS sites. Homogeneous time series with an average length of 14 years are available from 84 sites. Besides GNSS, there are 326 meteorological stations operated by the German weather service with data profiles spanning more than 60 years at a temporal rate of 1 hour . They provide surface measurements of temperature, pressure, water vapor pressure, precipitation, snow cover and other meteorological parameters for climate research. We also used the ERA-Interim reanalysis data with a spatial resolution of 40 km in longitude and 62 km in latitude and 6 hours temporal resolution. In this section, we briefly describe the methods for PWV determination and a comparison between the different data sets.

Based on the method of precise point positioning (Zumberge et al., 1997), GNSS observations are processed to produce site-specific atmospheric zenith total delay (ZTD). The ZTD is an estimate of the total propagation delay caused by the dry gases and water vapor of the atmosphere. Using meteorological data measured directly at the GNSS site or interpolated from the adjacent meteorological station, the zenith dry delay (ZDD) is calculated. For each GNSS site, the nearest meteorological station triangle is used to interpolate the measurements at that site (Gendt et al., 2004). The ZDD at the GNSS site is then calculated using the model of Saastamoinen (1983), i.e.,

$$ZDD = 0.002277\, D\,(P - 0.155471\, P_{wv}) \tag{1}$$

where the factor $D$ depends on the surface height ($h$) and the geographical latitude ($\phi$) such that

$$D = 1 + 0.0026\cos(2\phi) + 0.00028\, h \tag{2}$$

$P$ and $P_{wv}$ are the corresponding air pressure and the partial pressure of water vapor in hPa, respectively. $P_{wv}$ of water vapor is determined based on the relative humidity $rh$ and temperature $T$ from the following empirical formula:

$$P_{wv} = \frac{rh}{100}\cdot\exp\left(-37.2465 + 0.2131665T - 0.000256908T^2\right) \qquad \text{[hPa]} \tag{3}$$





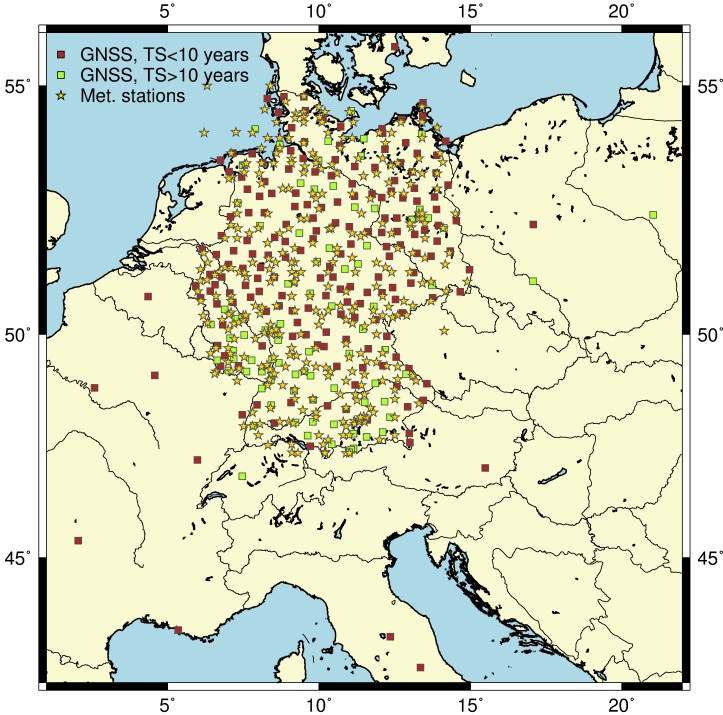

**Figure 1.** The location of the GNSS and meteorological sites distributed within the research region. 84 GNSS sites out of 278 have time series with an average length of 14 years.

The air pressure at the GNSS site $P$ in Eq. 1 is obtained by vertically interpolating the surface pressure $P_s$ using the barometric formula:

$$P = P_s \left( \frac{T_s - L(z - z_s)}{T_s} \right)^{gM/RL} \tag{4}$$

where $T_s$ is the air temperature at the meteorological station in [K], $z$ and $z_s$ are respectively the ellipsoidal altitude of the GNSS and meteorological station in [km], $L$ is the temperature lapse rate in [K/km], $R$ is the universal gas constant (8.31447 J/mol K), $M$ is the molar mass of Earth's air (0.0289644 kg/mol), and $g$ is the Earth's gravitational acceleration (9.80665 m/s$^2$). The temperature is related to the elevation change using the following linear regression:

$$T = T_s - L(z - z_s) \tag{5}$$

By analyzing temperature time series from a meteorological station on Zugspize ($\approx$ 2964 m above mean sea level (AMSL)) and another 7.61 km distant station at $\approx$ 719 m AMSL, we found the average value of $L$ for our research region to be 6.1 [K/km]. Once the ZDD is calculated, the zenith wet delay (ZWD) is obtained by:

$$ZWD = ZTD - ZDD \tag{6}$$





and it is converted into PWV using the empirical factor $\Pi$,

$$PWV = \Pi \cdot ZWD \tag{7}$$

We compared the PWV obtained from GNSS with ERA-Interim data. Figure 2 shows the results for three sites at different altitudes. Each cell of the ERA-Interim grid provides a mean PWV value of an area of about 40 km×60 km. The ECMWF provides a software to horizontally interpolate the current ERA-Interim grid at different locations of the GNSS stations as described in (Heise et al., 2009). For the sites located in flat terrain, the two data sets show strong correlation with a mean difference approaching zero and uncertainty values of of less than 0.4 mm (Table 1). For higher terrain (site 0285), however, a bias is observed between GNSS and ERA-Interim data. The reanalysis data might be responsible for the bias due to the averaging of PWV over large cells with highly variable surface topography, but GNSS might also induce some offset due to the shadowing effect in mountainous regions. We observed that the higher the GNSS antenna is located, the larger the bias.

For accurate determination of the PWV, it is required to have measurements of mainly air pressure and temperature at the GNSS sites or within a short spatial range. In the absence of meteorological measurements, would the interpolation of pressure and temperature from reanalysis data be a good replacement? To answer this question, we compared the PWV time series extracted from the ZTD by using both measurements at the meteorological stations and ERA-Interim data. The hereby measured pressure and temperature are horizontally interpolated to the GNSS site and then vertically interpolated to the altitude of GNSS the antenna to calculate the ZDD. The pressure and temperature are extrapolated at the altitude of the GNSS site using Equations 4 and 5, respectively. The ZWD is then extracted and converted into PWV. Figure 3 shows the scatterplots of PWV obtained using surface measurements and ERA-Interim data. We found that in regions of smooth topography, the ERA-Interim data and the measurements provide almost the same values of PWV. In regions of rough topography, however, the ERA-Interim data show slightly different results, which is mainly related to the pressure data as observed from Figure 3. The deviations between the measured pressure and the ERA-Interim pressure increases in mountainous regions, which affects the calculation of the ZDD and hence the obtained PWV.

| GNSS site | Mean [mm] | STD [mm] | Corr. Coef. |
|---|---|---|---|
| 0269 (Wertach, Bavaria) | -0.275 | 0.331 | 0.998 |
| 0522 (Pirmasens, Rhineland-Palatinate) | 0.205 | 0.189 | 0.999 |
| 0285 (Garmisch, Bavaria) | -0.598 | 0.492 | 0.983 |

**Table 1.** Comparison between PWV time series estimated from GNSS data using the PPP approach and ERA-Interim PWV.





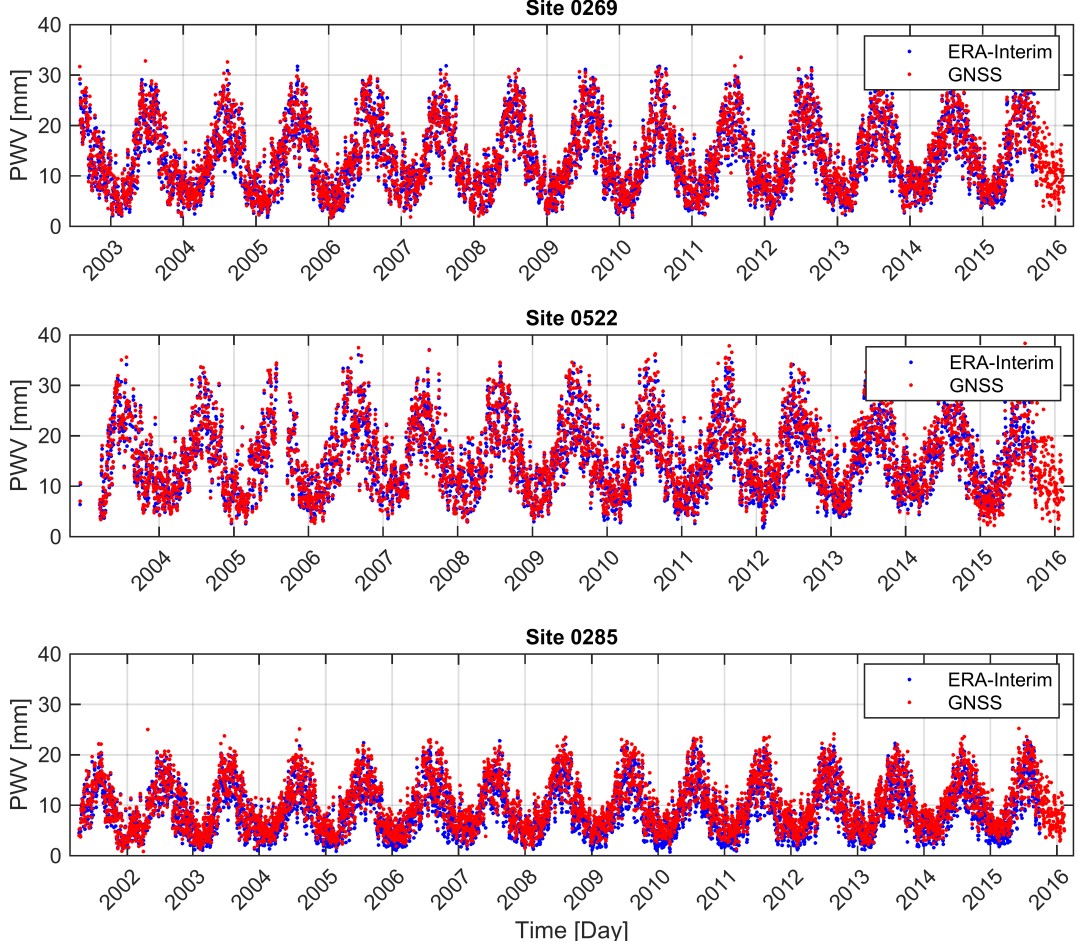

**Figure 2.** PWV estimated at three GNSS sites (site 0269 in Wertach, Germany at altitude of 907 m AMSL, site 0522 in Pirmasens, Germany at altitude of 399 m AMSL, and site 0285 in Garmisch, Germany at altitude of 1779 m AMSL) and the corresponding PWV from ERA-Interim.

An important factor for an accurate determination of PWV is the conversion factor $\Pi$, which should be calculated using measurements of surface temperature. Askne and Nordius (1987) determined the conversion factor $\Pi$ based on the weighted mean temperature of the atmosphere $T_m$ as follows:

$$\Pi = \frac{10^6}{\rho_w R_w \left( \dfrac{k_3}{T_m} + k_2' \right)} \tag{8}$$





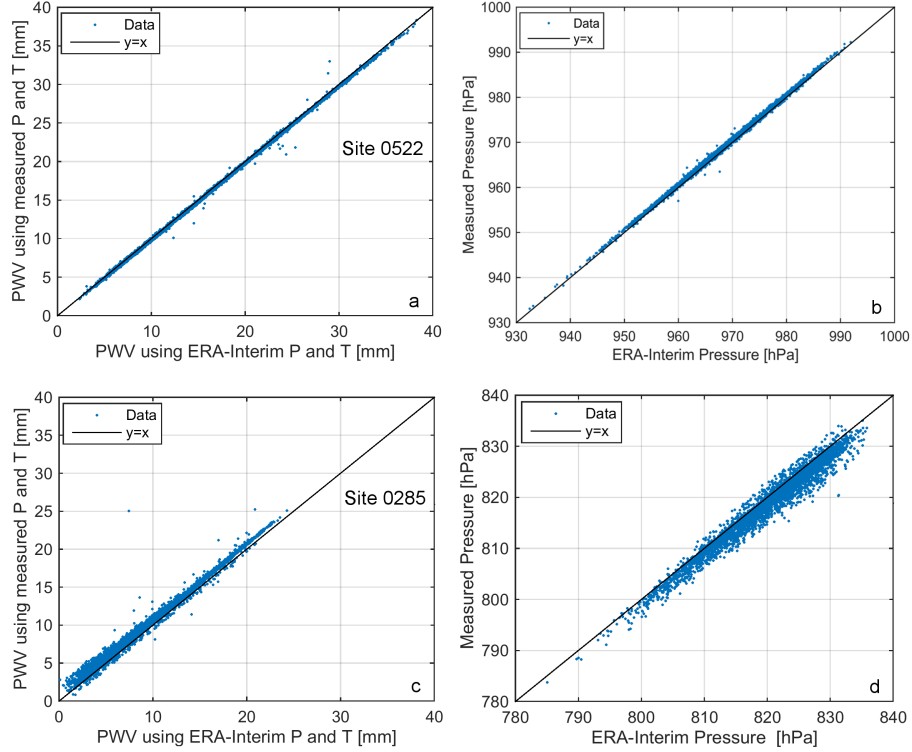

**Figure 3.** (a, b) show PWV determined using surface measurements of pressure and temperature against the PWV determined using simulations of surface pressure and temperature from ERA-Interim and the corresponding pressure values for the GNSS site 0522. Similarly in (c, d) for the GNSS site 0285.

where $\rho_w$ is the density of water and $R_w$ is the specific gas constant of water vapor [8.134 J/mol·K]. The values of the physical constants $k_3$ and $k_2^{'}$ are taken from Bevis et al. (1994), $T_m$ was given by Davis et al. (1985) as

$$T_m = \frac{\int_z \frac{P_{wv}}{T} dz}{\int_z \frac{P_{wv}}{T^2} dz} \qquad (9)$$

where $T$ is the air temperature. Davis et al. (1985) suggested the use of water vapor pressure and temperature profiles from

5 radiosondes; however, it is easier to get these profiles from atmospheric numerical models. In this work, we use the ERA-Interim data. $T_m$ can be well approximated based on air surface temperature by the following formula (Bevis et al., 1992):

$$T_m \approx 70.2 + 0.72 T_s \qquad (10)$$

$T_s$ is the surface temperature in [K]. For this research region, we compared $T_m$ obtained from both methods (9) and (10) as

10 shown by the scatterplot of Figure 4. The surface temperature and vertical profiles of water vapor pressure and temperature in Eq. 9, ERA-Interim data were employed. The difference between the $T_m$ calculated from both methods at the GNSS site 0522




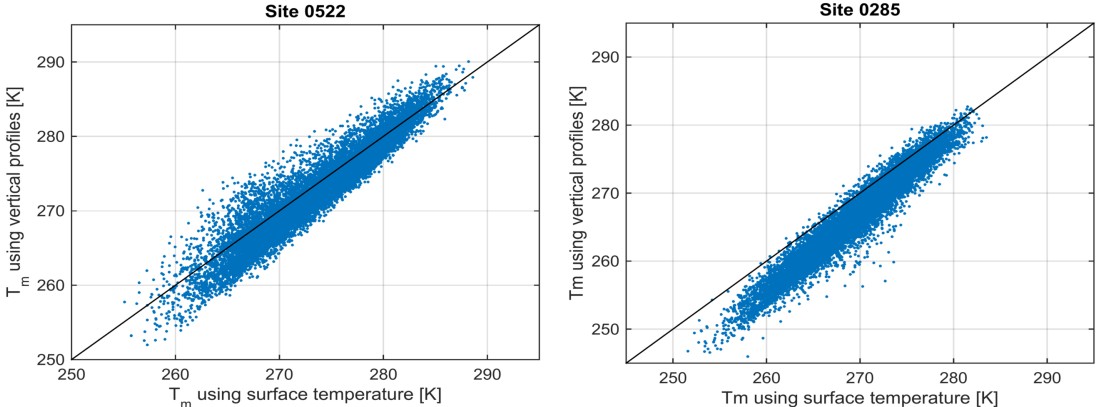

**Figure 4.** Mean atmospheric temperature, $T_m$ determined once using surface temperature and another using vertical atmospheric profiles from ERA-Interim at the sites 0522 (399 m AMSL) and 0285 (1779 m AMSL). The mean difference is 0.97 K for the first site and 3.02 for the second, and the STD is 2 K for the first and for the second 1.83 K.

(399 m AMSL) has a mean value of 0.97 K and a standard deviation (STD) of 2 K. Repeating the calculations for the site 0285 (1779 m AMSL), the mean difference increases to 3.02 K and the STD is 1.83 K. However, by computing the PWV using the two different values of $T_m$, the results show a mean difference of 0.048 mm. Hence, Eq. 10 will be used to calculate the mean atmospheric temperature since it only requires the measured surface temperature.

## 3  Determination of PWV based on surface meteorological measurements

It is not possible to accurately determine the total column water vapor directly using surface meteorological observations. However, it was shown in the 1960s that it is possible to approximate the atmospheric PWV from the dew point temperature, which is considered as an indicator of the amount of moisture in the air (Reitan, 1963). The dew point temperature in turn is determined based on the air temperature and relative humidity. Reitan (1963) presented a basic relationship between the mean monthly PWV and mean monthly surface dew point temperature by the following regression form:

$$PWV = \exp(bT_d + a) \tag{11}$$

where $PWV$ is in cm and $T_d$ is the dew point temperature in °F. $a$ and $b$ are estimated to have the values of -0.981 and 0.0341 (Reitan, 1963). The standard error in the PWV estimate was 0.18 cm. Following the same procedure, Bolsenga (1965) obtained slightly different estimates for $a$ and $b$ using hourly and mean daily observations. Smith (1966) obtained a similar regression equation with the coefficient $a$ not being constant. It rather depends on the vertical distribution of the atmospheric moisture, i.e.,

$$PWV = \exp(0.0393T_d + [0.1133 - \ln(\lambda + 1)]) \tag{12}$$





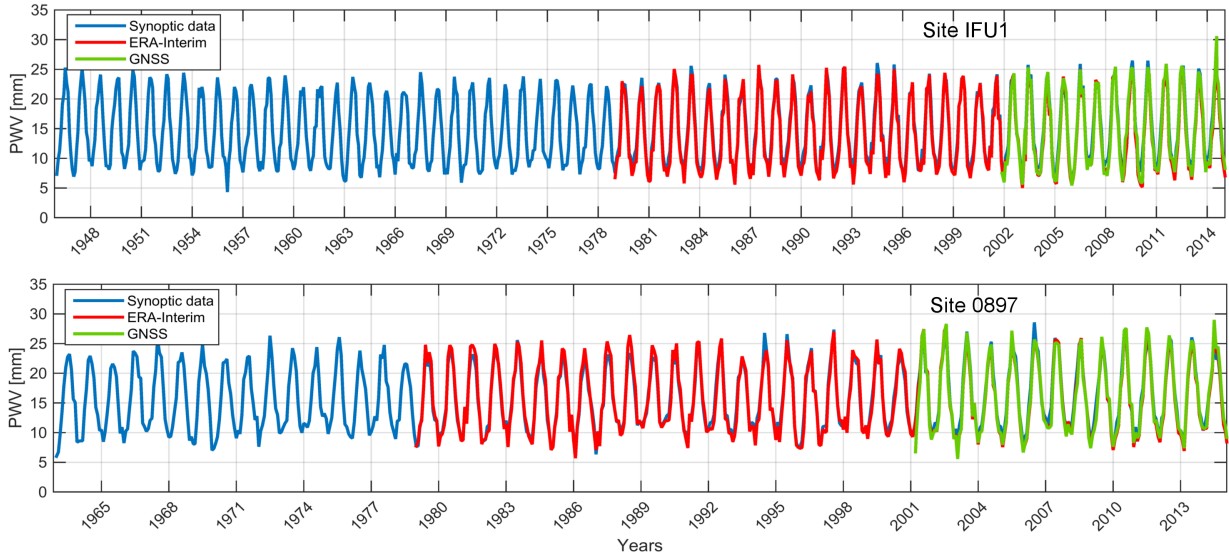

**Figure 5.** PWV time series obtained from GNSS observations, ERA-Interim reanalysis data and based on surface measurements of temperature and relative humidity at the sites 0897 (Berlin, Germany) and IFU1 (Garmisch, Germany).

with the value of $\lambda$ dependent on the site latitude and the season of year (Smith, 1966). The relative humidity is necessary to determine the dew point temperature $T_d$, which can be related as presented by Lawrence (2005) using the following formula:

$$T_d = T - \left( \frac{100 - rh}{5} \right) \tag{13}$$

where $rh$ is the surface relative humidity in percentage and $T$ is the surface air temperature. Both $T$ and $T_d$ are given in degrees Celsius. For our research region, using Eq. 11 and Eq. 12 shows marginal differences between the obtained PWV values. We compared the monthly mean PWV obtained from the three data sets at different GNSS sites, as shown in Figure 5. The results show strong correlation in the PWV values with uncertainty values below 1 mm (Table 2). After comparing the three data sets, we used them in the following to monitor the temporal evolution of PWV and other atmospheric variables.

| | Site 0897 (Berlin, Germany) | | | Site IFU1 (Garmisch, Germany) | | |
|---|---|---|---|---|---|---|
| Method | Mean [mm] | STD [mm] | Corr. Coef. | Mean [mm] | STD [mm] | Corr. Coef. |
| ERA-Interim/GNSS | -0.184 | 0.539 | 0.996 | -0.518 | 0.660 | 0.995 |
| ERA-Interim/Met. data | 0.193 | 0.975 | 0.987 | 0.825 | 0.767 | 0.992 |
| GNSS/Met. data | 0.447 | 1.053 | 0.987 | 0.623 | 0.849 | 0.992 |

**Table 2.** Comparison of PWV obtained from GNSS observations, ERA-Interim data, and surface meteorological measurements.





## 4 Decadal variability in time series of atmospheric variables

Econometricians developed reasonably simple models that are capable of interpreting, testing hypotheses, and forecasting economic data. The method was to decompose the time series into a trend, a seasonal, a cyclic, and an irregular component (Enders, 1995). The trend component represents the long-term behavior of the time series, while the seasonal and the cyclic components represent the regular and periodic movements. The irregular component is stochastic component. Time series of PWV and temperature, for example, have different temporal variations that can reasonably be modeled using these components. Here holds an additive model, such that the time series $y_t$ can be extended as:

$$y_t = T_t + S_t + I_t \tag{14}$$

where $T_t$ is a deterministic linear trend component with slow temporal variations, $S_t$ represents the seasonal component with known periodicity (e.g., 12 months for PWV and temperature), and $I_t$ represents the irregular (stationary) stochastic component with short temporal variations. We did not observe a regular signal that lasts longer than a year, so we excluded the cyclic component for the model. The presence of seasonality might mask the small changes in the linear trend. Therefore, for proper trend analysis, the seasonal component has to be estimated and removed from the time series, which is known by seasonal adjustment (Enders, 1995). The deseasonalized data are useful for extracting the long-term trend and exploring the irregular component of a time series.

The seasonal adjustment is applied as an iterative procedure as follows. To best estimate the seasonal component, the linear trend has first to be estimated and removed from the time series. There are different methods to estimate the trend, we applied a moving average filter with a window length of 1 year that is able to smooth out seasonal and irregular signals. We employ time series of PWV and temperature with daily values (the GNSS-based estimates of PWV have a temporal resolution of 15 minutes, but we average them to get mean daily values for climatological studies). The estimated trend is given by:

$$\hat{T}_t = \frac{y_{t-q} + y_{t-q+1} + \cdots + y_{t+q-1} + y_{t+q}}{d} \tag{15}$$

Since the time series are daily and the seasonal signal is annual, the value of $d$ is 365 and $q = (d-1)/2$. For $d = 366$, $q = d/2$ and the linear trend is estimated from:

$$\hat{T}_t = \frac{0.5y_{t-q} + y_{t-q+1} + \cdots + y_{t+q-1} + 0.5y_{t+q}}{d} \tag{16}$$

The estimated linear trend component is subtracted from the original time series and the detrended signal is used to estimate the seasonal component $\hat{S}_t$. We first estimate

$$w_t = \frac{1}{\text{number of summands}} \sum_{j=1}^{\frac{n-q-t}{d}} (y_{t+jd} - \hat{T}_{t+jd}) \tag{17}$$

with $n$ the number of data samples. Then

$$\hat{S}_t = w_t - \frac{1}{d} \sum_{k=1}^{d} w_k, \qquad t = 1, 2, \cdots, d \tag{18}$$





The second term of the above equation is added to estimate a seasonal signal with a zero mean. For an additive model, $\hat{S}_t$ should fluctuate around zero to avoid any influence from the linear trend. The estimated seasonal component is subtracted from the original time series to obtain a seasonally-adjusted time series $dy_t$, i.e.,

$$dy_t = y_t - \hat{S}_t \tag{19}$$

5   This signal is then used to obtain a second estimate of the linear trend $\hat{T}_t$, as described above. Figure 6 shows the trend, seasonal, and irregular components of PWV time series. We also applied a polynomial regression to estimate a fitting straight line $\hat{T}_t = a + b\,t$ and its slope from the deseasonalized signal.

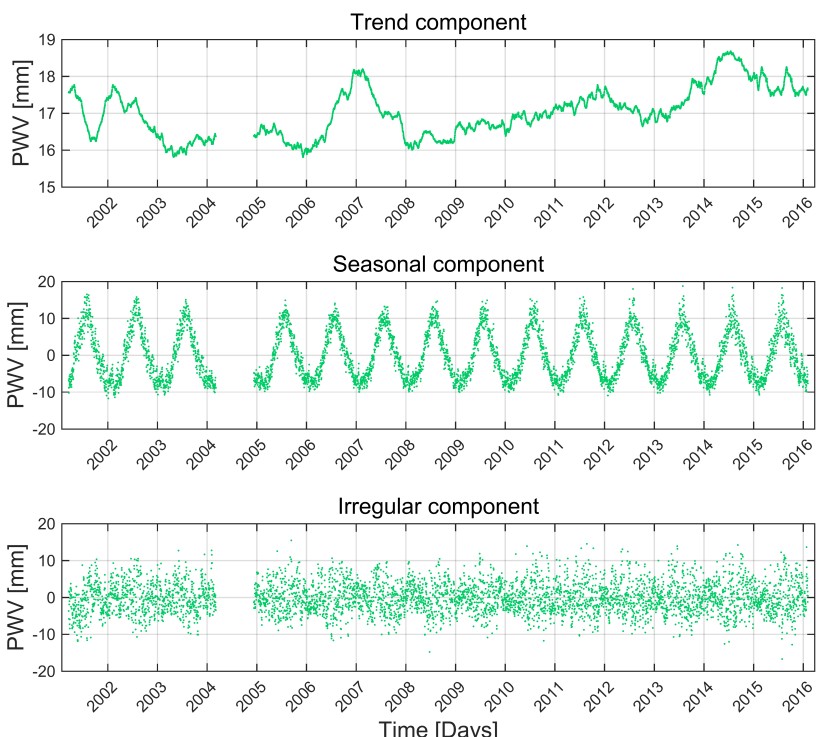

**Figure 6.** Trend, seasonal, and irregular components of PWV time series estimated from GNSS observations at the site 0896 (Berlin, Germany, 68.37 m AMSL).

We analyzed time series of PWV at the GNSS sites with time series of 14 years average length and the corresponding PWV from ERA-Interim and surface meteorological data. Figure 7 shows the smoothed, seasonally-adjusted PWV component and

10   the estimated linear trend at the GNSS site 0987 (Berlin, Germany, 78 m AMSL). The three data sets show good agreement in the PWV variability over time. The linear trend is estimated from ERA-Interim and dew-point-based PWV series for time windows of 30 years (climate normal). For last normal, the PWV level increases temporally as shown by the fitted straight line with a positive slope of 0.488 mm/decade, which is calculated using the dew-point-based PWV. By the analysis of temperature time series in a similar manner, the temperature time series show an upward trend of 0.33 °C/decade. The temporal variations



of the PWV correlates with the variations of temperature. Another example for the GNSS site IFU1 (Bavaria, Germany, 745 m AMSL) is shown in Figure 8. The PWV level tends to increase by 0.42 mm/decade and the temperature level increases with 0.417 °C/decade.

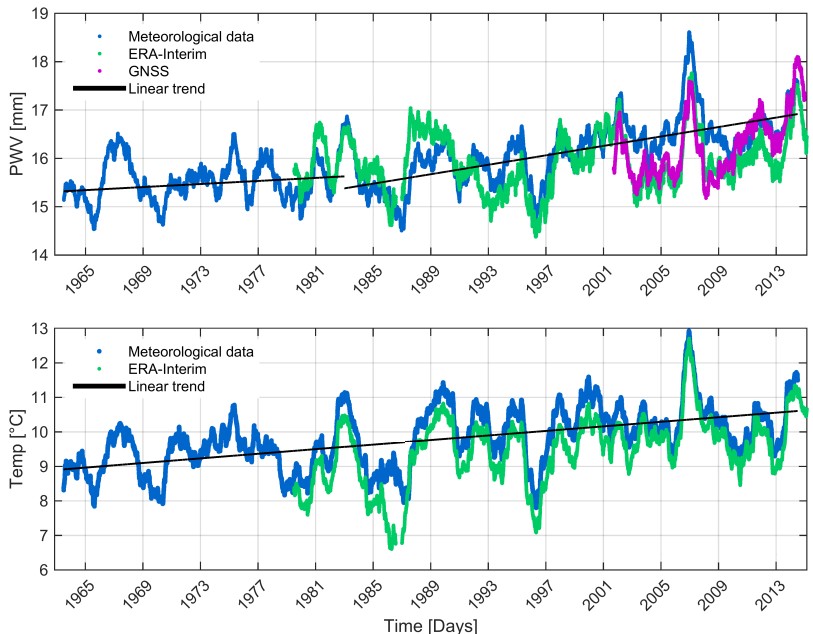

**Figure 7.** The upper graph shows the smoothed, seasonally-adjusted time series of PWV from GNSS, ERA-Interim, and meteorological data at the site 0897 (Berlin, Germany, 78 m AMSL). The lower graph shows the corresponding temperature time series. In black, the fitted straight lines over time windows of 30 years are shown. The PWV level increases by 0.488 mm/decade and the temperature level increases by 0.33 °C/decade.

For climate research, it is important to consider not only the change in PWV and temperature levels but also precipitation
5   and snow cover. We analyzed one data set for a site located on top of Zugspitze at altitude of 2963 m AMSL. Figure 9 displays the time series and the fitted straight line for dew-point-based PWV, temperature, precipitation, and snow cover. We observed an increase in the PWV and temperature levels with slopes of 0.11 mm/decade and 0.28 °C/decade, respectively. The snow cover, however, tends to decrease with a slope of -7.1 cm/decade. That is explained by the increase in temperature that causes a faster melting of snow or there might have been more precipitation in the form of rain than snow in the last 30 years. The
10   precipitation level increases by 0.11 mm/decade (Figure 9).

## 5   Conclusions

In this paper, we compared PWV time series obtained from GNSS, ERA-Interim, and meteorological data. The data sets show strong correlation with uncertainty values below 1 mm. By comparing the GNSS-based PWV with the ERA-Interim, the





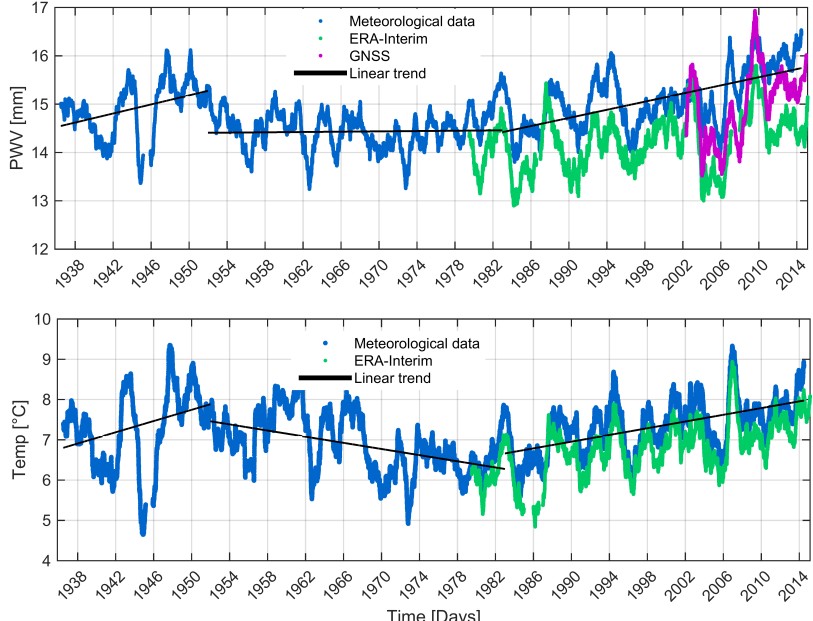

**Figure 8.** The upper plot shows the smoothed, seasonally-adjusted time series of PWV from GNSS, ERA-Interim, and meteorological data at the site IFU1 (Garmisch, Germany, 745 m AMSL). The lower graph shows the corresponding temperature time series. In black, the fitted straight lines over time windows of 30 years are shown. The PWV level increases by 0.42 mm/decade and the temperature level increases by 0.417 °C/decade.

results show strong agreement in flat terrain while a bias is observed in mountainous regions. In the absence of measurements of pressure and temperature, ERA-Interim data can be an appropriate replacement. In order to evaluate the temporal evolution of PWV and temperature, we modeled the time series with an additive model with a trend component, a seasonal component, and a stochastic irregular component. The time series are seasonally adjusted to remove the periodic signal and the trend component is analyzed. The time series from 3 data sets were used to estimate the trend of PWV, temperature, precipitation, and snow over 30 years. The results show a positive trend in the PWV content with an increase of 0.3–0.6 mm/decade. The increase in the PWV correlates with the temperature increase. In this paper, we monitored atmospheric variables at individual GNSS sites. In future work, we will monitor these variables at a regional scale to observe their temporal and spatial variability as well as the relation to climate impact.

10  *Acknowledgements.* The authors would like to thank the ECMWF for the ERA-Interim data. Thanks also go to the German weather service for providing us with hourly meteorological measurements.







**Figure 9.** Smoothed, seasonally-adjusted time series of PWV, temperature, snow cover, and precipitation and the fitting straight line for time windows of 30 years at the site Zugspitze (Garmisch, Germany, 2963 m AMSL). The slopes of the lines for the last climate normal are added to the figure.

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
