# Peer review of "Decadal variations in atmospheric water vapor time series estimated using ground-based GNSS"

_Atmospheric Measurement Techniques, 2016_

## Referee Comment (RC1) · Anonymous Referee #1 · 23 Jun 2016

Fadwa Alshawaf et al. described the comparison between precipitable water vapour (PWV) estimated with GNSS, ERA-Interim data and daily surface measurements of temperature and relative humidity. They examined on trends obtained by fitting a straight line into 30-years of seasonally-adjusted PWV. Their results show a positive trends for more than 60 GNSS stations from area of Germany and positive correlation between PWV and temperature recordings.

In my opinion the paper is worth publishing in Atmospheric Measurement Techniques, however, some issues are missing and few points should be raised for a discussion and expanded.

Major comments: 1. Page 3, line 10: "Homogenous time series with an average length

of 14 years are available from 84 sites". The total number of sites is 278. What with the rest of stations? Are the time series also homogenous? A comment on homogenization is needed here: give a total number of epochs applied, a maximum change in trend, a maximum change in standard deviation. Please, quantify a task of homogenization. 2. Page 5, line 10: "We observed that the higher the GNSS antenna is located, the larger the bias." How many stations are affected by this bias? Are the mean value and STD directly correlated with height? A comment on it is needed. 3. Page 8, fig. 4: site 0285: Where does the difference between Tm's below 260K come from? A comment on it should be added. 4. Page 10, line 3: You mentioned seasonal and cyclic component of ZTD data. What do you mean by cyclic? What is the difference between seasonal and cyclic? Why two terms should be mentioned? I would prefer to name cyclic as seasonal as well. 5. Page 10, line 10: "It represents the irregular (stationary) stochastic component with short temporal variations." The stationarity and short temporal variations are too optimistic to be assumed. In this way you input that the irregular component has no or little influence on determined parameters: trend and seasonal component. What if the stochastic component was correlated in time and in this way brought large uncertainties of trend and seasonals? 6. Page 11, fig. 6: The long-term variations you name as "trend component" may be related to noise model being far from white noise assumption you made. In fact, noise in PWV is close to autoregressive process. This is why the trend you estimate may be over/underestimated due to autoregressive trend and not necessarily real changes noticed in PWV data. Did you consider any other process being hidden in irregular component? A detailed comment on it should be added. 7. Page 13, line 1: "while a bias is observed in mountainous regions". Can you quantify this bias? 8. Page 14, fig. 9: Can you add the errors of estimated trends? It would help a reader to judge on its significance. 9. You show results for 3 different stations. Can you please add the statistics for all stations examined? It would give the overall view on stations and their (dis-)agreement with ERA-Interim and meteorological data. 10. Can you provide errors of all values of trend/slopes provided within the text. Now, a reader is not aware of significance of each value given.

Minor comments: 1. Time in figures is given in "years", not "days". 2. Page 1, line 10: "PWV trend component estimated from GNSS data strongly correlates with that..." give numbers to justify this "strong correlation". 3. Page 1, line 12: "0.3-0.6" an error must be added here. 4. Page 1, line 18: "a mount", change into "amount". 5. Page 5, line 7: double "of" 6. Page 7, fig. 3: the caption of bottom axis is not visible.

---

## Referee Comment (RC2) · Anonymous Referee #2 · 2 Jul 2016

Review of manuscript amt-2016-151 "Decadal variations in atmospheric water vapor time series estimated using ground-based GNSS" by Fadwa Alshawaf and co-authors.

General comments

The authors compare PWV from 3 data sources: GNSS, ERA-Interim and surface measurements of dew point temperature. They estimate trends in various parameters and find correlated positive trends in PWV and temperature in the last 30-years. The topic of this paper fits well the scope of AMT. However, I recommend major revision before publication.

My main objections are: - The use of dew point temperature as a proxy of PWV is

highly questionable when small variations and trends are to be extracted. To demonstrate the validity of this approach, a more comprehensive inter-comparison should be performed (not only for 2 sites), including daily data (since these are used to compute the trends) and also PWV trends. - Trend estimates are compared and interpreted but nothing is said about the significance of the values. It is mandatory to include uncertainty estimates and significance tests to conclude on the agreement of trend estimates and on the physical relationship between trends of different variables (e.g. PWV and temperature). - In order to strengthen the methodology and conclusions, data from more GNSS sites with homogeneous data should be included (only 3 sites are used in the inter-comparison and trend analysis whereas the authors claim they have 84 such sites).

The rationale and scientific questioning of the paper should be better introduced and data usage should be made accordingly (e.g. it is not clear why meteorological data back to 1900 are shown when only trends over the last 30-years are analysed).

Why are dew point temperature measurements used to extend the PWV series back into the past when ERA-Interim goes back to 1979 and other reanalyses exist which go further back in the past? Several global XXth century reanalyses have namely been released recently by ECMWF and NOAA.

It is not clear if the PWV comparisons in Section 2 are used to assess the accuracy of the GNSS PWV data or to highlight problems in the ERA-Interim data. A similar remark holds for the surface P and T measurements compared to ERA-Interim.

Nothing is said about the homogeneity of the meteorological data.

Specific comments

P2L15: The results of Bengtsson et al., 2004, are not used in a proper way. First, the trend value of +0.36mm/decade (global mean for the period 1979-2001) is deemed inconsistent by these authors who suggest it is an artefact caused by the changes in

the global observing system. They provide a more reasonable value after correction of +0.16mm/decade (global mean for the period 1958-2001). Second, it is mandatory to indicate the spatial and temporal domain when quoting such estimates because regional trends can be quite different (in sign and magnitude) from the global trend.

P2L18: The concluding sentence from this paragraph is wrong. The two quoted studies evidence strong limitations in the reanalysis data for characterizing long term trends and conclude on the necessity for better understanding and reducing the uncertainties in the long term trends from reanalyses.

P2L27-29: How can the current normal period be calculated until 2020 from observations? This slicing of time periods in the future makes only sense for model projections. When dealing with observations, the period of period should be present. Please correct the sentence accordingly.

P3L10: How is the homogeneity of the data from the 84 sites established or achieved? If any correction is applied to the data to homogenize them it should be explained here.

P3L11 & L20: Meteorological observations are used to calculate ZDD. The accuracy and homogeneity of these data and subsequent ZWD and PWV should be discussed.

P3L23: Equation (1): this formulation for ZDD, as an approximation of ZHD, is usually not used in the GNSS community. The commonly used formulation for ZHD is the one given by Davis et al., 1985, which based on Saastamoinen's earlier work of 1972/1973. Why is a different formulation used here? A consequence of using this formulation is place of ZHD is that the subsequent ZWD and PWV determined from equations (6) and (7) are not consistent with the commonly used formulations for these variables. Please justify your choice, assess the difference with standard formulations, or revise accordingly.

P5L3-10: The PWV data from GNSS and ERA-Interim are compared and it is concluded that the bias increases with height. Are the data corrected for height difference?

[Figure]

Please comment.

P5L10: what is the shadowing effect in mountainous regions? Please explain and quantify.

P5L20: To which extent is the bias at station 0285 explained by the pressure difference shown in Figure 3? Please provide an estimate of this effect.

P5 Table1: if altitude is a determining variable, please add it in the Table. Indicate also over which period the data are compared and at which temporal resolution (monthly?).

P8L4: section 2 lacks a conclusion on the PWV, surface pressure and Tm comparisons.

P9L3: equation (13) is a very poor formula to convert rh to Td as emphasized by Lawrence (2005). Either account for the related uncertainty and propagate it to the PWV and trend estimates or use a more elaborate formula from Lawrence (2005).

P9 Table2: specify the temporal resolution (monthly?)

P10L1-3: Why citing statistical methods used in econometrics? A reference from the climate literature would be more in the scope of this paper.

P10: Equations (15) and (16): how are the trends calculated near the edges of the time series?

P11L15 & P12L2: compare the PWV – T relationship to the Clausius-Clapeyron equation.

Add uncertainty estimates to the trends.

Include regression results for more sites to assess the spatial variability.

Why is only the last 30-year period analysed? The change compared to previous periods is also of interest.

Figure 7 & 8: there are quite large biases between the different datasets. Please comment and assess the impact on the trend estimates.

P12L8: it is a very quick and hazardous conclusion that the observed temperature increase (0.28°C/decade) causes faster melting of snow or that precipitation is more in form of rain. Please justify or revise.

Revise the conclusions (section 5) accordingly.

Technical comments

Please put all the figures at the end of the manuscript (see the AMT author guidelines for more details). Indicate the period of comparisons and temporal resolution of the data in all figures presenting data. Figure 3: wrong labelling: (a,c) PWV and (b,d) surface pressure. Add station ID in the title of plots. Add station altitude in the captions. Figure 3 & 4: Add statistics of differences in the plot (mean, std.dev., correlation). P1L18: PWV is the amount of water *that would* result from condensing... P1L23: define GPS acronym P7L1: The numeric value for Rw (specific gas constant of water vapour) is wrong. P7L6: specify if model-level or pressure-level data are used. P8L3: for which site is the difference of 0.048 mm found? Give the numbers for both sites.

---

## Referee Comment (RC3) · Anonymous Referee #3 · 7 Jul 2016

Comments on « Decadal Variations in atmospheric water vapor time series estimated using ground-based GNSS »

By F. Alshawaf et al.

The authors used several datasets, including ground-based GNSS, to estimate variations and trends of precipitable water vapor over three stations in Germany. They first compare different methodologies to get PWV and then extract different factors which generate the observed variability : the trend, the seasonal cycle and the short-term variability. The study deserves to be published, however several aspects should be improved before it can be accepted for publication. I thus recommend major revision following the following comments :

- introduction : the objectives of the paper should be better explained.

- The authors say they analyzed time series of PWV at the 84 GNSS sites but they only show the results at two sites and do not discuss the results of the global analysis. Please add a more regional discussion of the results (the authors propose a regional analysis in future work but I think part of this analysis should be inlcuded in this paper).

- The relationship between PWV and temperature trends shoudl be better assessed : does it follow Clausius-Clapeyron relationship ? If not explain.

- We do not know if the computed trends are significant or not. Errors are missing.

- According to equation 15 or 16, it is not clear how you compute trends at the beginning and end of the time series.

- Figs. 7 and 8 : there are some differences between the three methods that are discussed. The black line is the fitting straight line of which dataset ?

- Tm shows strong differences when using surface temperature or vertical profiles of ERA-interim. The authors do not explain the huge bias at site 0285. They finally conclude that they can use equation 10 because the mean difference they generate in the computation of PWV is weak. However, I would like to see a scatter plots of these differences because a mean differnce is not enough to convince the reader it does not impact the value.

- Another issue on the methodology is the use of a constant (in space and time if I understood well) lapse rate of temperature. Isn't it a big approximation ?

- The part with snow and precipitation is too poor. Either you bette analyse the role of snow and precipitation (other sites, trend in the occurrence of T over 0°…), either you remove it from this study.

---

## Author Comment (AC1) · 17 Aug 2016

Response of the authors:

The authors would like to thank the editor and the reviewers for the time investigated to review this paper. We addressed the suggested points. The current version contains all changes according to the reviewer suggestions.

| Referee#1 | Response of the authors |
|---|---|
| Major comments: 1. Page 3, line 10: "Homogenous time series with an average length of 14 years are available from 84 sites". The total number of sites is 278. What with the rest of stations? Are the time series also homogenous? A comment on homogenization is needed here: give a total number of epochs applied, a maximum change in trend, a maximum change in standard deviation. Please, quantify a task of homogenization. | Of the 278 sites, the time series at 84 sites are longer than 10 years. Since the length of the time series is critical for climate studies, we involved only these 84 sites in the trend analysis. Because these time series are still not sufficiently long for climate studies, we used other data sets. We agree with the reviewer that homogenization is a great topic for this work. We have here a paper on homogenization for global network Ning, T.,Wickert, J., Deng, Z., Heise, S., Dick, G., Vey, S., and Schöne, T.: Homogenized time series of the atmospheric water vapor content obtained from the GNSS reprocessed data, Journal of Climate, 2016. No critical change points were observed for the sites in Germany; however, we are currently working on this specific area to evaluate the ZTD and PWV products. This will be published in another paper. |
| 2.Page 5, line 10: "We observed that the higher the GNSS antenna is located, the larger the bias." How many stations are affected by this bias? Are the mean value and STD directly correlated with height? A comment on it is needed. | For most regions, the topography is rather flat. This effect is observed in the Alps region, where an ERA-Interim cell over 70 km in each direction averages the topography around the Zugspitze and in here, we observe the bias between GNSS and ERA-Interim. Within that cell, there exist 3 GNSS sites and based on them we cannot judge the dependence of the mean and STD on the height, but apparently for these 3 sites the mean did. |
| 3. Page 8, fig. 4: site 0285: Where does the difference between Tm's below 260K come from? A comment on it should be added. | Not only surface pressure grids are inaccurate in mountainous regions (Figure3-d), but also pressure profiles because of the coarse grid of ERA-Interim. Also, the temperature profiles have inaccuracies even though less than that for the pressure. By using the integration in Eq. 9, the accumulated error in the calculated Tm will be higher; and the bias between this Tm and the Tm calculated using only the surface temperature will increase, as observed from the right plot in Figure 4. Comment added in text. |
| 4. Page 10, line 3: You mentioned seasonal and cyclic component of ZTD data. What do you mean by cyclic? What is the difference between seasonal | We agree with the reviewer on this point. However, we used a model developed by econometricians, who defined a cyclic component as a seasonal |

| | |
|---|---|
| and
cyclic? Why two terms should be mentioned? I would prefer to name cyclic as seasonal as well. | signal of period of several years. We presented the term for completeness, but we only stayed with the seasonal component. |
| 5. Page 10, line 10: "It represents the irregular (stationary) stochastic component with short temporal variations." The stationarity and short temporal variations are too optimistic to be assumed. In this way you input that the irregular component has no or little influence on determined parameters: trend and seasonal component.
What if the stochastic component was correlated in time and in this way brought large uncertainties of trend and seasonals? | According to our tests, the stochastic signal might show temporal correlation up to 7 days but not longer. For the trend analysis, a temporal filter of 1 year is used to remove its effect. And the seasonal component is estimated first and removed. This is all done iteratively until the three components are distinguished.
If that does not answer the question, we will be thankful if you suggest a reference to read. |
| 6. Page 11, fig. 6: The long-term variations you name as "trend component" may be related to noise model being far from white noise assumption you made. In fact, noise in PWV is close to autoregressive process.
This is why the trend you estimate may be over/underestimated due to autoregressive trend and not necessarily real changes noticed in PWV data. Did you consider any other process being hidden in irregular component? A detailed comment on it should be added. | No we did not, but according to this comment we will invest more analysis on this component. Thank you. |
| 7. Page 13, line 1: "while a bias is observed in mountainous regions". Can you quantify this bias? | Added to text |
| 8. Page 14, fig. 9: Can you add the errors of estimated trends?
It would help a reader to judge on its significance. | Added |
| 9. You show results for 3 different stations. Can you please add the statistics for all stations examined? It would give the overall view on stations and their (dis-)agreement with ERA-Interim and meteorological data. | We are still working on the entire network. It is early to make conclusions, but a preliminary result of the change at all stations is shown in the figure below. |
| 10. Can you provide errors of all values of trend/slopes provided within the text.
Now, a reader is not aware of significance of each value given. | Added |

| | |
|---|---|
| Minor comments: 1. Time in figures is given in "years", not "days". 2. | Done |
| Page 1, line 10: "PWV trend component estimated from GNSS data strongly correlates with that :::" | Added |
| give numbers to justify this "strong correlation". 3. Page 1, line 12: "0.3-0.6" an error must be added here. | Added |
| 4. Page 1, line 18: "a mount", change into "amount". | Done |
| 5. Page 5, line 7: double "of" 6. Page 7, fig. 3: the caption of bottom axis is not visible. | Done |

[Figure]

The change of PWV and temperature over the previous 30 years

---

## Author Comment (AC2) · 18 Aug 2016

Response of the authors:

The authors would like to thank the editor and the reviewers for the time investigated to review this paper. We addressed the suggested points. The current version contains all changes according to the reviewer suggestions.

| Referee#2 | Response of the authors |
|---|---|
| My main objections are: - The use of dew point temperature as a proxy of PWV is highly questionable when small variations and trends are to be extracted. To demonstrate the validity of this approach, a more comprehensive inter-comparison should be performed (not only for 2 sites), including daily data (since these are used to compute the trends) and also PWV trends. - Trend estimates are compared and interpreted but nothing is said about the significance of the values. It is mandatory to include uncertainty estimates and significance tests to conclude on the agreement of trend estimates and on the physical relationship between trends of different variables (e.g. PWV and temperature). | We also thought that using these data might be critical; however, when comparing the PWV time series obtained based on the dew point temperature, we found a small bias to the PWV measured for example radiosondes. Of course we need to test more sites. This paper presents the concept and preliminary results (title is changed) and we are working on the whole network and will in future work be more able to give more specific conclusions. |
| In order to strengthen the methodology and conclusions, data from more GNSS sites with homogeneous data should be included (only 3 sites are used in the inter-comparison and trend analysis whereas the authors claim they have 84 such sites). | That is right, that you. We need to analyze more sites, which what we are currently doing; however, it is quite early to make conclusions about the entire region. Fig. 1 below shows a first results for all available sites. |
| The rationale and scientific questioning of the paper should be better introduced and data usage should be made accordingly (e.g. it is not clear why meteorological data back to 1900 are shown when only trends over the last 30-years are analysed). | We want to consider longer time periods than 30 years; however, according to the availability of the GNSS data we compared the last 30 years of the time series. |
| Why are dew point temperature measurements used to extend the PWV series back into the past when ERA-Interim goes back to 1979 and other reanalyses exist which go further back in the past? Several global XXth century reanalyses have namely been released recently by ECMWF and NOAA. | That is because we want to rely on measurements and not the model data in their current spatial resolution. And because we think there is a large potential in these homogeneous measurements for the analysis of atmospheric variables. Of course more effort has to be put on evaluating the quality of the data. |
| It is not clear if the PWV comparisons in Section 2 are used to assess the accuracy of the GNSS PWV data or to highlight problems in the ERA-Interim data. A similar remark holds for the surface P and T measurements compared to ERA-Interim. | Since the GNSS PWV time series are not long enough for trend analysis, it was necessary to find another data set, which was the ERA-Interim. So, we compare the GNSS and ERA-Interim to show that it is reasonable to use PWV time series from ERA-Interim for PWV trend analysis as well as |

| | temperature. Added to text. |
|---|---|
| Nothing is said about the homogeneity of the meteorological data. | Yes, this is unintentionally missing! We added that in text. They are provided by the German weather service for climate studies and they are homogeneous. |
| Specific comments | |
| P2L15: The results of Bengtsson et al., 2004, are not used in a proper way. First, the trend value of +0.36mm/decade (global mean for the period 1979-2001) is deemed inconsistent by these authors who suggest it is an artefact caused by the changes in the global observing system. They provide a more reasonable value after correction of +0.16mm/decade (global mean for the period 1958-2001). Second, it is mandatory to indicate the spatial and temporal domain when quoting such estimates because regional trends can be quite different (in sign and magnitude) from the global trend. P2L18: The concluding sentence from this paragraph is wrong. The two quoted studies evidence strong limitations in the reanalysis data for characterizing long term trends and conclude on the necessity for better understanding and reducing the uncertainties in the long term trends from reanalyzes. | Text is modified |
| P2L27-29: How can the current normal period be calculated until 2020 from observations? This slicing of time periods in the future makes only sense for model projections. When dealing with observations, the period of period should be present. Please correct the sentence accordingly. | According to the climatologists, the current normal goes from 1991-2020. We, however, are still in 2016, so we defined the interval from 1984-2014 depending on the availability of archive data. It was given in text (now modified). |
| P3L10: How is the homogeneity of the data from the 84 sites established or achieved? If any correction is applied to the data to homogenize them it should be explained here. | We had this paper on homogeneity of global GNSS sites. Ning, T.,Wickert, J., Deng, Z., Heise, S., Dick, G., Vey, S., and Schöne, T.: Homogenized time series of the atmospheric water vapor content obtained from the GNSS reprocessed data, Journal of Climate, 2016. For the sites in Germany, we did not detect any change points in the analyzed sites. We are working on a paper that goes into details for all 278 sites in the research region. We calculated the difference between GNSS time series of PWV and radiosonde |

| | and model data to detect sudden disconnect in the time series. |
|---|---|
| P3L11 & L20: Meteorological observations are used to calculate ZDD. The accuracy and homogeneity of these data and subsequent ZWD and PWV should be discussed. | That is a good point. We did not describe the GPS data processing in this paper and for more details, we refer to the following papers: Gendt, G., Dick, G., Reigber, C., Tomassini, M., Liu, Y., and Ramatschi, M.: Near real time GPS water vapor monitoring for numerical weather prediction in Germany, J. Meteor. Soc. Japan, 82, 361–370, 2004.

Bender, Michael, et al. "Development of a GNSS water vapour tomography system using algebraic reconstruction techniques." Advances in Space Research 47.10 (2011): 1704-1720.

If not measured at the GPS site, the pressure and temperature are interpolated from 3 neighboring stations and are accepted with an error of $\pm 1$ hPa and $\pm 1$ K. |
| P3L23: Equation (1): this formulation for ZDD, as an approximation of ZHD, is usually not used in the GNSS community. The commonly used formulation for ZHD is the one given by Davis et al., 1985, which based on Saastamoinen's earlier work of 1972/1973. Why is a different formulation used here? A consequence of using this formulation is place of ZHD is that the subsequent ZWD and PWV determined from equations (6) and (7) are not consistent with the commonly used formulations for these variables. Please justify your choice, assess the difference with standard formulations, or revise accordingly. | Indeed there have been different works in the GNSS community that aimed at improving the estimation of PWV from GNSS and this formula was used, see for example,
1. Troller, Marc. *GPS based determination of the integrated and spatially distributed water vapor in the troposphere*. Vol. 67. 2004.
2. Luo, X., B. Heck, and J. L. Awange. "Improving the estimation of zenith dry tropospheric delays using regional surface meteorological data." *Advances in Space Research* 52.12 (2013): 2204-2214.
3. Alshawaf, F., T. Fuhrmann, A. Knöpfler, X. Luo, M. Mayer, S. Hinz, B. Heck (2015). Accurate estimation of atmospheric water vapor using GNSS observations and surface meteorological data. Transactions on Geoscience and Remote Sensing. 53 (7), pp. 3764–3771, IEEE Journals & Magazines.

However, in this work, we used the traditional Saastamoinen model and the text has been modified. |
| P5L3-10: The PWV data from GNSS and ERA-Interim are compared and it is concluded that the bias increases with height. Are the data corrected for height difference? Please comment. | This conclusion is made for the mountainous region where a cell of ERA-Interim data of 70 km in each direction averages the topography around the Zugspitze. So this sentence is not precise for other area with a rather smooth topography. It is modified |

| | in text to avoid misunderstanding. And yes, we interpolate the ERA-Interim data at the GNSS site for the sake of comparison. |
|---|---|
| P5L10: what is the shadowing effect in mountainous regions? Please explain and quantify. | Due to the presence of mountains, the visibility of satellites might be limited. Also, there might be multipath effects in the observed signal. This will have an impact on the estimated tropospheric parameters. This is added to text. |
| P5L20: To which extent is the bias at station 0285 explained by the pressure difference shown in Figure 3? Please provide an estimate of this effect. | According to what we understand from the questions: The site 0285 is located within the ERA-Interim cell that contains the Zugspitze and a variable surface elevation over 70 km × 70 km. Therefore, the atmospheric variables in this cell are inaccurate, e.g., pressure, temperature, and PWV. To determine the PWV at the GNSS site, we use the measured pressure at from meteorological stations. How the PWV is provided in the ECMWF model or how the pressure might affect it is beyond the scope of this paper. Did we get your point correctly? |
| P5 Table1: if altitude is a determining variable, please add it in the Table. Indicate also over which period the data are compared and at which temporal resolution (monthly?). | The site altitudes are added to the table, too. In the figure the time period is given on the x-axis at a temporal resolution of one day (x-label). |
| P8L4: section 2 lacks a conclusion on the PWV, surface pressure and Tm comparisons. | Text added |
| P9L3: equation (13) is a very poor formula to convert rh to Td as emphasized by Lawrence (2005). Either account for the related uncertainty and propagate it to the PWV and trend estimates or use a more elaborate formula from Lawrence (2005). | That is right, thank you. Using this formula to obtained the Td results in 0.38 mm mean difference in PWV. We replaced it with the most accurate formula in Lawrence 2005. |
| P9 Table2: specify the temporal resolution (monthly?) | Yes, monthly. Added to caption. |
| P10L1-3: Why citing statistical methods used in econometrics? A reference from the climate literature would be more in the scope of this paper. | Thank you. However, Science is indivisible, so we think what important is to add the right citation. |
| P10: Equations (15) and (16): how are the trends calculated near the edges of the time series? | We estimate one trend for the entire interval without putting a difference near the edges. |

| | |
|---|---|
| P11L15 & P12L2: compare the PWV – T relationship to the Clausius-Clapeyron equation. | Please refer to Fig. 2 and table 1). We checked the rate of change of the PWV, vapor pressure, and saturation pressure, first using the total values and then looking only at the trend (seasonal and irregular component removed). For the first case, the rate of change in the saturation pressure follows Clausius-Clapeyron relation and the two sites show roughly consistent values. (For Garmisch, the values tend to be smaller, we think because it is located higher). If we look only at the trend component, however, we could not make conclusions, the sites behave differently. If the analysis we did does not answer the question of the reviewer, we would kindly ask him/her to recommend a reference we can refer to do the required analysis. |
| Add uncertainty estimates to the trends. | Added |
| Include regression results for more sites to assess the spatial variability. | We agree with the reviewer on this point. However, we do not think that adding more sites will give enough indication about the spatial variability of the trends. Therefore, we make use of the spatial properties of atmospheric variables to provide 2D trend estimates over time. This topic has to be well-described and it will be submitted as an independent paper. |
| Why is only the last 30-year period analysed? The change compared to previous periods is also of interest. | Yes, it is of interest; however, we wanted to compare the three data sets and the period of GNSS as well as Era-Interim are limited. By the end of this year the ERA-Interim data will be available for the entire century, so we can do more analysis. Moreover, we considered the shortest time period the climatologists suggest. |
| Figure 7 & 8: there are quite large biases between the different datasets. Please comment and assess the impact on the trend estimates. | As we can see from table 2, the bias for both sites does not exceed 1 mm. It increases for the site IFU1 between ERA-Interim and the other 2 data sets because as we explained above the ERA-Interim data for this mountainous cell are not adequately accurate even if interpolated and downscaled. The bias, however, has little to do the slope of the regression. |
| P12L8: it is a very quick and hazardous conclusion that the observed temperature increase (0.28 C/decade) causes faster melting of snow or that precipitation is more in form of rain. Please justify or revise. Revise the conclusions (section 5) accordingly. | This part is removed from this paper |

| | |
|---|---|
| Technical comments
Please put all the figures at the end of the manuscript (see the AMT author guidelines for more details). | Done |
| Indicate the period of comparisons and temporal resolution of the data in all figures presenting data. | Done |
| Figure 3: wrong labelling: (a,c) PWV and (b,d) surface pressure. Add station ID in the title of plots. Add station altitude in the captions. | Modified |
| Figure 3 & 4: Add statistics of differences in the plot (mean, std.dev., correlation). | Added |
| P1L18: PWV is the amount of water *that would* result from condensing
:::
P1L23: define GPS | Done |
| | Done |
| acronym P7L1: The numeric value for Rw (specific gas constant of water vapour) is wrong. | Corrected |
| P7L6: specify if model-level or pressure-level data are used. | Text modified |
| P8L3: for which site is the difference of 0.048 mm found? Give the numbers for both sites. | Text modified |

[Figure]

[Figure]

Fig.1: The change of PWV and temperature over the previous 30 years

[Figure]

Fig.2: PWV and water vapor pressure against temperature

Table 1: Comparison of the rate of change of PWV and water vapor pressure for sites Berlin and Garmisch

|  | Site Berlin [%/K] | | Site Garmisch [%/K] | |
| --- | --- | --- | --- | --- |
|  | Total | Trend only | Total | Trend only |
| Saturation vapor pressure | 7.5 | 4.7 | 7.3 | 3.7 |
| Vapor pressure | 6.7 | 8.9 | 6.6 | 4 |
| Column water vapor | 5.9 | 9.3 | 5.6 | 6.5 |

---

## Author Comment (AC3) · 18 Aug 2016

Response of the authors:

The authors would like to thank the editor and the reviewers for the time investigated to review this paper. We addressed the suggested points. The current version contains all changes according to the reviewer suggestions.

| Referee#3 | Response of the authors |
|---|---|
| The objectives of the paper should be better explained.
-
The authors say
they analyzed time series of PWV at the 84 GNSS sites but they only
show the results at two
sites and do not discuss the results of the global analysis
Please add a more regional discussion of the results (the authors propose a regional
analysis in future work but I think part of this analysis should be inlcuded in this paper). | Abstract and introduction are modified.

We are still working on the entire network. It is early to make conclusions, but a preliminary results of the change at all stations is shown in the Fig.1 (below table). |
| The relationship between PWV and temperature trends shoudl be better assessed :does it follow Clausius-Clapeyron relationship?
If not explain. | Please refer to Fig. 2 and table 1). We checked the rate of change of the PWV, vapor pressure, and saturation pressure, first using the total values and then looking only at the trend (seasonal and irregular component removed). For the first case, the rate of change in the saturation pressure follows Clausius-Clapeyron relation and the two sites show roughly consistent values. (For Garmisch, the values tend to be smaller, we think because it is located higher). If we look only at the trend component, however, we could not make conclusions, the sites behave differently.
If the analysis we did does not answer the question of the reviewer, we would kindly ask him/her to recommend a reference we can refer to do the required analysis. |
| We do not know if the computed trends are significant or not. Errors are missing.
- | Added |
| According to equation 15 or 16, it is not clear how you compute trends at the beginning and end of the time series.
- | They are added. |
| Figs 7 and 8 :there are some differences between the three methods that are  discussed. The black line is the fitting straight line of which dataset ?
- | We estimate one trend for the entire interval without putting a difference near the edges. |
| Tm shows strong differences when using surface temperature or vertical profiles of
ERA-interim. The authors do not explain the huge | Text modified with more explanation.
Please see Fig. 3 (below). |

| | |
|---|---|
| bias at site 0285. They finally conclude that they can use equation 10 because the mean difference they generate
in the computation of PWV is weak. However, I would like to see a scatter plots of these differences because a mean differnce is not enough to convince the reader it does not impact the value.
-
Another issue on the methodology is the use of a constant (in space and time if I Understood well) lapse rate of temperature. Isn't it a big approximation ?
-
The part with snow and precipitation is too poor. Either you bette analyse the role of
snow and precipitation (other sites, trend in the occurrence of T over 0°...), either you
remove it from this study. |

Yes we agree with the reviewer that the lapse rate should not be constant in space and time and it will be adapted in future work.

This part is removed from this paper |

[Figure]

Fig.1: The change of PWV and temperature over the previous 30 years

[Figure]

Fig.2: PWV and water vapor pressure against temperature

Table 1: Comparison of the rate of change of PWV and water vapor pressure for sites Berlin and Garmisch

|  | Site Berlin [%/K] | | Site Garmisch [%/K] | |
|---|---|---|---|---|
|  | Total | Trend only | Total | Trend only |
| Saturation vapor pressure | 7.5 | 4.7 | 7.3 | 3.7 |
| Vapor pressure | 6.7 | 8.9 | 6.6 | 4 |
| Column water vapor | 5.9 | 9.3 | 5.6 | 6.5 |

[Figure]

Fig.3: $\Pi$ factor calculated using the Tm in Fig. 4 (in the paper) and the corresponding ZWD (Eq. 7)